# Burnout Syndrome and COVID-19 Lockdown: Research on Residential Care Workers Who Assume Parental Roles with Youths

**DOI:** 10.3390/ijerph192316320

**Published:** 2022-12-06

**Authors:** Laura Ferro, Marina Cariello, Alessandra Colombesi, Alberto Segantini, Eleonora Centonze, Giorgia Baccini, Stefania Cristofanelli

**Affiliations:** 1Department of Psychology, Faculty of Psychology, University of Valle d’Aosta, 11100 Aosta, Italy; 2TIARE’, Association for Mental Health, 10125 Turin, Italy

**Keywords:** COVID-19, burnout syndrome, health care professionals, parental burnout, youths

## Abstract

Healthcare professionals are at higher risk of developing and experiencing burnout. Parents may also suffer from prolonged stressful conditions that lead to physical and emotional exhaustion. Residential youth care workers assume a caregiving role that can lead to persistent stressful conditions that affect their relationship with the youth. In addition, the COVID-19 lockdown has had a negative impact on both the organization and the work, as well as on the lifestyle of workers and minors. In fact, during the pandemic, contact with families was not possible due to restrictions and this increased the need for caregivers to assume a parental role. This research aims to examine the risk of burnout in a sample of 75 healthcare professionals working with youths and the association with psychological traits. Then, we aim to evaluate these aspects during the COVID-19 lockdown The measurements, conducted in both February 2019 and April 2021, included six questionnaires: MBI to assess burnout, TAS_20 to explore alexithymic traits, COPE_NVI to assess coping strategies, FDS_R to quantify frustration intolerance at work, IRI for empathy, and FFMQ to investigate awareness and emotional regulation. Our sample shows a medium-high risk of developing burnout, which worsened during the pandemic. A worsening of emotional skills, paralleled by a greater empathic investment required by the emergency situation, and an assumed parental role is observable. Coping strategies correlate with burnout risk, as avoidance strategies were strongly associated with emotional exhaustion. These findings suggest an urgent need to develop targeted and timely interventions for healthcare professionals in order to prevent long-term consequences.

## 1. Introduction

The term burnout means to burn something to the point of exhaustion, to the greatest power available. In the workplace, people can respond to high levels of chronic stress and trigger pathological behaviors that can lead to the development of “burnout syndrome”. This definition was introduced by Freudenberg [1] and further developed by Maslach [2,3,4,5]. Burnout has physical and psychological symptoms and involves a strong sense of frustration. Burnout syndrome is characterized by feelings of emotional exhaustion, a lack of personal accomplishment, and depersonalization [6,7]. Emotional exhaustion is the main characteristic associated with stress and is defined as a lack of physical and mental strength required to perform daily work tasks. The lack of personal accomplishment and satisfaction leads to negative attitudes toward oneself, low self-esteem, dissatisfaction, and feelings of professional failure. Depersonalization involves the assumption of aloofness, cynicism, and a negative attitude toward other people [6,7]. These three components are assessed by the Maslach Burnout Inventory (MBI) [8]. Burnout syndrome can affect individuals of any age and occupational category, but is more common in workers who interact with others and are involved in professional relationships such as helping, supporting, and educating [9,10]. In particular, healthcare professionals are at increased risk of experiencing depression and anxiety in response to stress and traumatic situations [11]. Education professionals, such as teachers, also exhibit high levels of burnout, low scores on perceived self-efficacy, low job satisfaction, and low levels of professional engagement [12]. In addition, for parents, persistent stress that severely and chronically overstretches parental resources can also lead to physical and emotional exhaustion, and thus burnout [13,14]. Stressful conditions may involve involvement in daily activities such as time management, household tasks, specific disease states, or physical, mental, and behavioral frailty, as well as specific crises, e.g., during adolescence. The three dimensions of parental burnout are emotional fatigue, emotional disengagement, feelings of failure, and ineffectiveness in relation to parenting practices [13,15,16]. To conserve remaining energy, parents tend to distance themselves from their children and lose the enjoyment of activities, making their role unbearable. The magnitude and frequency of these symptoms can be used to determine whether they are experiencing normal stress or burnout [14]. 

In residential youth care, the helping relationship between healthcare professionals and minors contains specific parental elements that can significantly influence the risk of burnout. The function of residential treatment has developed heterogeneously in the international context. In the Mediterranean context, and particularly in Spain and Italy, social pedagogy is firmly established as a discipline and profession; in residential care, there is a high level of professional qualification, theoretical models, and specific working tools that generate and maintain quality in residential child protection services [17]. Nevertheless, residential care can provide an overarching experience of safety, resilience, and a sense of belonging for minors at risk [18]. They are places that integrate and temporarily replace parental functions when they are impaired and maladaptive. The minors concerned are in fact in a psychosocial risk situation that makes it necessary to remove them from their context of origin and to maintain relations with this milieu as much as possible. Residential treatment is based on the construction of a therapeutic environment in which the different professional groups of personnel work according to specific functions, but in a common clinical health context. In Italy, local administrations define forms and methods of organization and supervision of residential treatment for the protection of children and adolescents. They also define the specific training and professional requirements for professionals, who are also selected with regard to their interpersonal skills, healthy personality profile, willingness to listen, and receptivity. The main professionals in residential treatment are the director in charge, a team of psychologists and psychotherapists who perform the functions of health coordinators, supervisors, individual and family psychotherapists, group activity psychologists, psychodiagnostics, professional educators, nurses, and healthcare assistants. Shealy [19] defined professionals who work in such settings as “therapeutic parents”, that is, healthcare professionals who perform therapeutics but with parent-like tasks, such as supervision and teaching daily living skills. Anglin [20] also referred to a “home-like” environment in the residential care context, where professionals are called on to respond as effectively as possible to the needs expressed by minors [21]. The residential care professional is the primary caregiver for the minor and for the family, taking direct responsibility for the minor and acting as a “surrogate parent” [22]. Youths tend to use the healthcare professionals involved in residential care as a secure base where there must be a balance between empathy and emotional support and collaboration on tasks and goals [23]. In residential treatment, the relationship is a source of therapeutic reparation for the harm and deprivation suffered in the family context. The healthcare professional is present in the minor’s daily life; professionals establish a close relationship with youths. The residential worker becomes the caregiver and is actively protective in daily interactions [24]. The focus of the work is direct and continuous participation in the “habitat” of the minor he is caring for on a daily basis [25]. He becomes a familiar figure who helps to create meaning in an atmosphere of knowledge and reliability; it is an authentic, mutual, respectful, and asymmetrical relationship [26,27]. In residential treatment, professionals, who as caregivers have closer and daily contact with minors, are called to invest physically, emotionally, and spiritually in the physical and mental health of minors to promote their appropriate development and well-being. Residential caregivers become the most important adult figures because they deal with the needs of minors on a daily basis; daily interactions are at the heart of residential care [21,28]. Given these realities, it should be clear that these caregivers have an enormous influence and responsibility for the youth in their care [18,24,28,29]. Residential treatment can be very stimulating and stressful due to the commitment and specificity of the work context [25,30].

The profession is one of the most difficult and emotionally stressful in the personal services field. Professionals seem to experience aspects of burnout in particular ways compared to other human services professions precisely because of the emotional commitment that caring for minors requires [30,31]. Low commitment and poor job satisfaction can lead to less empathy and availability and can affect the quality of the relationship [28,32]. Poor organizational climate can affect the ability of these caregivers to adequately represent their role and lead to increased turnover and depersonalization rates [28,33]. Risk factors for the occurrence of burnout among these professionals also include a lack of support when demands are too high compared to available resources [30]. Social support from colleagues, supervisors, friends, and family appears to be a protective factor against work-related stress and the risk of burnout [34]. It is, therefore, necessary to promote support services for residential care professionals to adaptively cope with job demands through supervision, training, and support programs [29]. However, research also describes that higher levels of stress may be associated with greater commitment, dedication, and concern for the care and well-being of youths [28,32]. It appears that these workers, although mentally and physically exhausted, still feel obligated and committed to successfully completing their work [25]. However, there is a lack of analysis and reflection in the literature on the stressful conditions experienced by youth care professionals In particular, there is a lack of data on the impact of the COVID-19 pandemic on these workers.

Therefore, the purpose of this analysis is to examine specific aspects of burnout in a sample of healthcare professionals working as caregivers in residential care and protection facilities. The study identifies the specific dimensions of burnout using the Maslach Burnout Inventory (MBI) [8]. It also examines the ability to use emotional resources as a source of individual support, empathic investment, and coping strategies needed to manage external stress demands that are important for the risk of the onset of burnout. Subsequently, starting from the impact that the health emergency that COVID-19 had on the well-being of the whole population, we wanted to explore any changes in our sample. Indeed, the literature has examined how factors such as social distancing, lockdown, periods of isolation, fear of illness, and economic, relational, and social consequences have affected working conditions [35]. In particular, during an epidemic, factors such as the increase in working hours, the loss of balance of well-being, and the lack of support in the professional and family environment can cause severe emotional and psychophysical stress, up to the appearance of burnout symptoms [36]. In particular, a significant number of healthcare workers showed nervousness, irritability, frustration, discouragement and anxiety, depressive and post-traumatic symptoms, sleep disturbances, the impairment of the quality of external relationships and, in general, quality of life [37,38]. The level of emotional exhaustion was significantly higher than in the pre-pandemic period; however, there was still a proportion of work-related satisfaction [39]. However, there is little information in the literature on strategies to prevent stress and affect the well-being of healthcare professionals [11].

Parental stress was also influenced by COVID-19, increasing concern for children’s health and social isolation, the duration and quality of online parenting, and the ability to provide age-appropriate information to children [40]. Levels of parental burnout have increased substantially in many countries around the world [41]. High levels of burnout affect parenting practices and reduce the use of positive parenting strategies [42]. Previous research has shown that parents with higher levels of burnout are more likely to experience physical and mental exhaustion, disengagement from their children, and feelings of incompetence about their role [43]. The psychological impact of the pandemic on parental stress and the occurrence of burnout-related symptoms, such as exhaustion, has also been associated with less positive child behaviors, confirming the strong link between parental and child well-being [44]. In contrast, other analyses show significant levels of emotional exhaustion only in certain risk situations [13], such as motherhood, single parenthood, younger children, children with special needs and large numbers of children, and disadvantaged economic circumstances [13,41,44].

Compared to what we know, there is currently only one study on the impact of COVID-19 on healthcare professionals directly involved in the care of youths in residential treatment [45] but to date, the risk of burnout among these professionals during the health emergency has not been studied. Therefore, the second objective of our study is to examine burnout risk during the acute phase of the epidemic, when the context of personal and professional life has changed dramatically. Lack of support and encouragement, such as the possibility of collaboration with the public health emergency service, the extracurricular activities of the school, such as sports facilities, and the internal and external laboratories, may have further influenced the risk of occurrence of burnout. The goal is to help researchers and stakeholders develop targeted mental health coping strategies to prevent long-term causes. 

## 2. Materials and Methods

### 2.1. Study Design

We conducted the study using an anonymous Google Forms survey sent to healthcare professionals working in residential centers for youth with psycho-social problems The first data collection took place in February 2019, with the aim of investigating the psychological distress of residential care workers. After the occurrence of the COVID-19 emergency, we decided to repeat the assessment. The second data collection took place in April 2021. In fact, in full lockdown, the residential care workers had to work in a context with greater challenges, such as the obligation to work with masks, sudden lack of habitual social entertainment, full health institutions, and shortage of staff to deal with other emergencies.

The sample was recruited by sending an email to the coordinators of residential centers for minors throughout the Italian territory. Once consent was received, the test battery was sent. A single institutional email address received all anonymous responses.

### 2.2. Participants

During the first survey, a total of 100 healthcare professionals completed the entire survey, but only 75 of the first sample responded in April 2021.

The inclusion criteria for the study were being a healthcare professional working in a residential center for youth with psycho-social problems and being in close contact with the minor. Only workers hired for more than three months were included, so that the caregiver–youth relationship had time to establish itself. In particular, a daily report was required, as was typical for those who work in residential structures and shifts at different times of the day. Therefore, we excluded the coordinators of the structures, neuropsychiatrists, therapists, social assistants, trainees, and other healthcare figures responsible for managing activities inside and outside the structure (pet therapists, etc.). So, we have included educators, nurses, psychiatric rehabilitation technicians, and healthcare assistants. 

Participants were, on average, 40.22 years old (SD 10.6). As described in Table 1, most of them were women all working in residential youth care and located in northern Italy. Most of them had a three-year degree (42.1%) and were married (40%). Most participants played some sports (occasional = 47.4%; regular = 42.1%).

### 2.3. Instruments

In February 2019, all participants received an anamnestic questionnaire that included all sociodemographic information and occupational details. In both February 2019 and April 2021, they answered six self-report questionnaires: the Maslach Burnout Inventory (MBI), the Toronto Alexithymia Scale (TAS-20), the Coping Orientation to Problems Experienced (COPE IV), the Frustration Discomfort Scale (FDS-R), the Interpersonal Reactivity Index (IRI), and the Five Facet Mindfulness Questionnaire (FFMQ). In April 2021, participants were given an additional survey (created by the authors) that examined changes after the pandemic experience. 

#### 2.3.1. Anamnestic Form

The form was preceded by a note about the purpose of the study and information about consent and privacy with consent for anonymous data processing. The anamnestic form consisted of a collection of sociodemographic and occupational data. Specifically, the anamnestic data collected refer to gender, age, marital status, education, physical activity, years of seniority, distance from the workplace, type of work (shift work, employment contract), number and age of hosted minors, professional and psychological support, and job satisfaction.

#### 2.3.2. COVID-19 Survey

The additional survey was created by the authors and administered during the follow-up (April 2021). We assessed the occurrence of quarantine, changes in workload, aggression by minors, and emotional experiences related to pandemic constraints. The emotional states assessed were both positive and negative. Positives were confidence, solidarity, a sense of usefulness, and efficacy. Negatives were fear, distress, worry, anger, anxiety, and confusion 

#### 2.3.3. Self-Report Questionnaires

##### Maslach Burnout Inventory

The Maslach Burnout Inventory (MBI) highlighted the aspects related to work stress through its three dimensions: emotional exhaustion (EE), depersonalization (DP), and personal accomplishment (PA) [46]. The Italian version was used [47]. It consists of twenty-two items, and the worker answers each item on a six-point Likert scale, expressing the frequency with which each emotional state has been experienced during the last week (0 = never, 6 = every day). The Italian version deviates slightly from the psychometric values of the original American version. Cronbach’s alpha coefficient is 0.87 for the EE subscale, 0.76 for the PA subscale, and 0.68 for the DP subscale.

##### Toronto Alexithymia Scale

Emotional functioning was assessed by the Toronto Alexithymia Scale (TAS-20), which is composed of three factors: difficulty identifying feelings, difficulty in describing feelings, and externally-oriented thinking [48]. Test scoring includes the total score and measurement of three components of alexithymia: difficulty identifying feelings (F1), difficulty describing feelings to others (F2), and externally oriented thinking (F3). The twenty items are evaluated using a five-point Likert scale, starting with 1 (strongly disagree) to 5 (strongly agree). The Italian version was administered. Bressi and colleagues [49] demonstrated factorial validity, internal consistency (0.75 for total score, 0.77 for F1, 0.67 for F2, 0.52 for F3), and high test-retest reliability. 

##### Interpersonal Reactivity Index

The Interpersonal Reactivity Index (IRI) was used to better explore emotivity by assessing empathy. This self-report questionnaire consists of twenty-eight items with a four-point Likert scale (1 = Never to 4 = Always true). It has four subscales, namely perspective-taking, fantasy, empathic concern, and personal distress [50]. The Italian version confirms the four factors and has a sufficiently adequate internal consistency. For the Fantasy factor, the alpha value is 0.74; for Perspective Taking it is 0.64, for Empathic Concern it is 0.63, for Personal Distress it is 0.64. For the overall scale, the alpha is 0.75 [51].

##### Five Facet Mindfulness Questionnaire

The Five Facet Mindfulness Questionnaire is a measure of psychological well-being related to the concept of mindfulness. It is a multifactorial scale, consisting of thirty-nine items and five components: observing, describing, acting with awareness, not judging the inner experience, and nonreactivity to the inner experience [52]. The Italian version of FFMQ [53] has a similar factor structure to the original English version and has good to excellent internal consistency as a whole (alpha = 0.86) with sub-scale consistency ranging from 0.65 to 0.81 (Observing: 0.79, Describing: 0.89, Acting with awareness: 0.86, Not judging: 0.86, Nonreactivity: 0.74).

##### Coping Orientation to Problems Experienced

Coping strategies were assessed using the Coping Orientation to Problems Experienced (COPE IV). It includes the assessment of five dimensions: social support, avoidance strategies, positive attitude, problem-solving, and turning to religion [54]. The Coping Orientation to Problems Experienced—New Italian Version (COPE-NVI) [55] is a sixty-item questionnaire with a four-point Likert scale ranging from 1 (I typically do not do it) to 4 (I almost always do it). Internal consistency for the five dimensions assessed in the Italian version is good: 0.91 for social support, 0.70 for avoidance strategies, 0.76 for positive attitudes, 0.83 for problem orientation, and 0.85 for transcendent orientation.

##### Frustration Discomfort Scale

The Frustration Discomfort Scale (FDS_R) is a multidimensional measure of frustration intolerance. The dimensions assessed are discomfort intolerance (DI), achievement (A), entitlement (E), and emotional intolerance (EI) [56]. The first factor reflects the belief that thoughts and feelings associated with emotional distress are intolerable. The entitlement factor represents the belief that one’s desires must be met and that other people should indulge and not frustrate these desires. The discomfort intolerance factor refers to demands that life should be easy, comfortable, and free of problems. The last factor reflects perfectionistic achievement beliefs associated with frustration intolerance. The questionnaire has 28 items. In the Italian version, factorial validity and internal consistency are confirmed: 0.92 for the full scale, 0.87 for DI, 0.76 for A, 0.82 for E, and 0.64 for EI [57].

### 2.4. Statistical Analysis

Data were analyzed using IBM SPSS Statistics version 27 software (SPSS Inc., Chicago, IL, USA). First, we performed descriptive statistical analyses. The Wilcoxon test for paired samples was used for comparison between the first and second assessments. The association between psychological and occupational factors and the risk of developing burnout resulted from the calculation of Spearman’s rank correlation coefficient.

## 3. Results

### 3.1. Descriptive Statistics

Table 2 and Table 3 show descriptive statistics of the variables examined in the anamnestic form and the COVID-19 survey. All information refers only to the 75 participants who answered in April 2021. 

Participants had an average professional seniority of 10.6 years (SD = 7.8).

Most employment contracts are permanent (86.6%) and full-time (77.3%). The sample is homogeneous in terms of the age of the minors, who are both prepubescents (9–14 years old = 50.7%) and adolescents (15–18 years old = 49.3%). Most of the healthcare professionals in our sample work in a residential structure where 8–10 minors live (77.3%). A total of 60.5% of the sample receives psychological support weekly, but there is a minority (2.6%) who never receive any type of support. Moreover, satisfaction with the organizational policy is greater than with the profit. In fact, overall,57.9% of the participants are satisfied with the organization (good satisfaction = 47.4%; very good satisfaction = 10.5%), compared to only 21% who are satisfied with the profit (good s. = 18.4%; very good s. = 2.6%).

The answers to our COVID-19 survey show an overall increase in work hours and physical and verbal aggression by minors during the COVID-19 emergency. Changes in relationships between colleagues are heterogeneous, with improvement in 26.7% of cases and deterioration in 40% of cases. Almost half of the sample (44.7%) experienced many quarantine episodes. Most participants reported feeling more useful (64.5%), supportive (64.5%), and concerned (63.1%).

### 3.2. Burnout Assessment 

Figure 1 shows the percentage of subjects at high, medium, and low risk of developing burnout with respect to the dimension of emotional exhaustion, as measured in February 2019. Overall, almost half (48%) had a medium-low risk of burnout. The assessment that took place during the COVID-19 lockdown (2021), on the other hand, reveals variations in the percentages of burnout risk. High-risk workers are increasing (47%) and medium-risk (28%) and low-risk (25%) are decreasing. Figure 2 and Figure 3 show the risk percentages referred to as personal accomplishment and depersonalization.

In April 2021, all scores on the dimensions of the MBI exceeded normative cut-offs [8]. These observations were confirmed by the Wilcoxon test (see Table 4). No gender difference was found.

As we can see in Table 4, the results of the Wilcoxon test show that there are statistically significant differences in the risk of developing burnout. Indeed, there was a significant increase in emotional exhaustion (EE) (*z* value = 5.82; *p* < 0.001). Depersonalization (DP) also changed, but to a lesser extent than EE (*z* value = 4.39; *p* < 0.001). However, the pandemic also appears to have a strong impact on personal accomplishment (PA), which decreased significantly in our sample (*z* value = −4.78; *p* < 0.001). 

### 3.3. Psychological Assessment 

Table 5 shows the results of the psychological assessment.

The Wilcoxon analysis revealed significant changes in other psychological traits.

We observed significant differences in the use of coping strategies (total score COPE: *z* value = 4,2; *p* < 0.001) in the degree of frustration intolerance (total score FDS: *z* value = 3.55; *p* < 0.001), emotional functioning (total score TAS: *z* value = 5.08; *p* < 0.001), and in overall aspects of the FFMQ (total score FFMQ: *z* value = −2.86; *p* = 0.004). Significant differences were found in some dimensions of empathy. More specifically, empathic concern (*z* value = 3.02; *p* = 0.003), perspective taking (*z* value = 3.87; *p* < 0.001), and personal distress (*z* value = −4.75; *p* < 0.001). However, these changes were ambivalent. On the one hand, there was a general deterioration in all dimensions of emotional functioning, with the appearance of greater use of externally oriented thinking and greater difficulty in describing and identifying feelings. Consistent with this finding, the emotional skills of describing, acting, and non-judging (explored by FFMQ) were less observed. Discomfort and emotional intolerance were also more prevalent at follow-up. Regarding the use of coping strategies, we found greater employment of avoidance strategies after the COVID-19 emergency. On the other hand, the longitudinal analysis also showed that participants used perspective-taking and empathic concern, with unexpectedly lower levels of personal distress than in the pre-emergency period.

### 3.4. Correlation Analysis

No strong significant correlation between burnout risk and psychological traits was found. Nevertheless, in April 2021, avoidance coping strategies were associated with high levels of emotional exhaustion (Rho = 0.62; *p* < 0.001). No correlation between occupational factors and burnout was found. Additional trends can be seen in Appendix A.

## 4. Discussion

The aim of this study was to investigate the presence of burnout risk in a specific sample composed of healthcare professionals working in residential structures for youth. As is well known, the care performed in residential structures contains elements typical of parental relationships. Indeed, working with minors requires being responsive to their emotional needs, which implies a significant emotional investment on the part of the workers [21]. Moreover, the literature confirms that burnout syndrome is not limited to work, but also affects the parenting function [16,58,59].

First, specific aspects of job burnout were examined using the Maslach Burnout Inventory (MBI) [8], namely perceptions of emotional exhaustion, lack of job fulfillment, and depersonalization [6,7]. Emotional exhaustion is the most important dimension related to stress and is defined as the lack of physical and mental strength needed to complete daily work tasks. The lack of personal accomplishment and satisfaction leads to a negative attitude toward oneself, low self-esteem, dissatisfaction, and a sense of failure on the job. Depersonalization involves the assumption of distance, cynicism, and a negative attitude toward other people [6,7]. At the first evaluation, which took place in February 2019, about half of the sample had a moderate to high risk of developing burnout. This dataare consistent with the literature, which reports average burnout levels that tend to be higher among healthcare workers [47]. On the other hand, the results of the study have some peculiarities. The first assessment shows a risk of burnout associated with the dimension of personal satisfaction to a greater extent than with the other two dimensions. These data are consistent with other studies showing the role of personal accomplishment in increasing work stress [30]. Moreover, the average age of our sample is 40 years with a standard deviation of 10.6 years, suggesting that there are young workers. The same is true for the number of years worked: an average of 10 years with a DS of 7.8. Studies suggest that workers with fewer years of service are at greater risk of experiencing low personal satisfaction. The hypothesis is that there is an “adjustment period” in which the worker is more vulnerable [30,60].

The second assessment allowed us to highlight other aspects related to the way health workers respond to stress. During the pandemic period, the level of burnout increased significantly among both caregivers and parents, especially in relation to the dimension of emotional exhaustion [11,13,35,36,37,38,39,40,41,44].

However, to our knowledge, there have been studies of burnout symptoms during and after the pandemic in caregivers and parents, but no studies to date have examined the role of COVID-19 on caregiver well-being

Consistent with the literature [39], our study showed a level of emotional exhaustion that was significantly higher than the levels found before the COVID-19 pandemic; more specifically, in our sample, the mean value of the level of emotional exhaustion was even higher than the data in the literature [39]. This may suggest that emotional skills are more exposed to exhaustion in healthcare professionals working with adolescents with emotional and behavioral difficulties. These results are also consistent with the extent of emotional exhaustion in parents in most countries of the world [41,44], although some results have shown that the specific dimension of emotional exhaustion is not affected, except in special risk conditions [13].

Regarding personal accomplishment, our sample showed a greater deterioration in the level of job satisfaction and gratification after the lockdown than reported in the literature [38]. Again, the results are related to the lower satisfaction and sense of parental failure experienced by parents during the COVID-19 pandemic. Previous research has found that parents who experience elevated levels of burnout are more likely to experience physical and mental exhaustion, emotional distancing from their children, and feelings of incompetence in their parenting role [43]. In addition, the literature suggests that for parents experiencing burnout, there is a discrepancy between expectations of the parenting role and perceptions of whether or not those expectations have been met [43]. It is possible that parents who face highly stressful situations experience a lack of resources that leads to unconscious parenting and, therefore, creates a sense of being less competent [57]. Compared to occupational burnout, our results are partially contradictory, as the literature shows that the COVID-19 pandemic affects emotional exhaustion in healthcare professionals, but not personal accomplishment [39]. The fact that the personal accomplishments dimension was also significantly impaired in our study suggests that working with minors may be associated with particular emotional and professional investments. Indeed, working as an aide and as a caregiver to minors may further impair the ability to maintain a solid sense of self-esteem and a positive perception of one’s own abilities and role in an external situation of high stress, such as during a pandemic.

The third dimension of job burnout, depersonalization, had smaller effects compared to the other two dimensions. However, the variation is still significant and consistent with the literature on COVID-19 and job burnout [39]. Depersonalization is an aspect that is far removed from the parenting context. Even when parents appear very exhausted, they do not “depersonalize” their children, but distance themselves from the sources of exhaustion. More specifically, parents of children with externalized disorders are emotionally rather than physically distanced, continue to attend to practical matters, and are less emotionally involved [16]. Indeed, in the Parental Burnout Assessment (PBA), the depersonalization subscale was replaced by the emotional distancing scale.

However, the component of emotional distancing from sources of exhaustion related to parental burnout may have been identified in our sample through the analysis of coping strategies. Measurements showed a significant increase in avoidance strategies. These findings suggest the hypothesis that healthcare professionals who work with minors, as in our sample, act with an emotional detachment under conditions of physical and mental exhaustion. This attitude may resemble what happens in the relationship between parents and children [43]. Healthcare professionals, similar to parents, may distance themselves and avoid dealing with adolescents in order to save their remaining energy [37].

In this direction, the emotional skills of professionals were also analyzed and showed a general deterioration of skills, especially the competence to recognize emotions. These results could imply that the caregiver’s function in a highly stressful situation, such as a pandemic, hinders emotional self-regulation. This may be particularly the case when fear, anger, and confusion are present. Emotion regulation skills may influence the caregiving experience by mitigating the effects of negative emotions in the minors they care for.

We also examined frustration tolerance. The literature states that higher levels of frustration tolerance are associated with a lower index of psychological well-being [61]. In our study, the most significant data were the variation in the dimension of emotional intolerance. Consistent with studies including parents [14], we hypothesize that burnout symptoms may play a role in avoidance and disengagement among healthcare professionals working with minors, to the point of poor tolerance of their role as caregivers.

We examined possible resources as protective factors for the occurrence of burnout, but a correlational analysis revealed no significant strong interactions.

In our sample, empathy concern and perspective-taking increased significantly. It is possible that the COVID-19 pandemic increased the attitude of feeling the experience of others as part of one’s own experience. However, high levels of empathic involvement, along with difficulty in recognizing one’s own emotions and a low tolerance for emotions, may have further impaired the ability to regulate emotional distress, especially during a time when relationships and social support were impaired because of the limitations. These observations also suggest the possibility that the immediate responses to the pandemic event, while necessary to functionally cope with the emergency, could lead to long-term difficulties [62].

In addition, the correlational analysis showed that greater reliance on avoidance strategies correlated with greater emotional exhaustion, but only in the second evaluation (during the COVID-19 emergency). Similarly, we found a low correlation between avoidance strategies and personal accomplishment. The use of avoidance strategies appears to be associated with lower job satisfaction.

The study has some limitations. First, due to the number and characteristics of the sample, it was not possible to collect representative data. In addition, due to the impossibility of using the Parental Burnout Assessment, we were unable to examine the stress associated with educational work with minors. The literature [16] suggests that the structure and content of parental burnout is somewhat different from the perspectives of job burnout, both at the theoretical and practical levels. Finally, we cannot rule out the possibility that other variables not measured in the study influenced burnout risk.

Future perspectives concern the possibility of expanding the sample to the whole Italian territory. A more detailed analysis of the typical characteristics of work with minors could provide further suggestions.

## 5. Conclusions

The study is part of a broader work to analyze the professional well-being of residential youth care workers. This occupational category has the distinction of defining itself as a healthcare profession that simultaneously performs a caregiving function and replaces the parental one. The literature shows that the professions most likely to develop burnout symptoms are precisely those in which relationships are helpful and supportive; at the same time, research shows that there are situations of parental burnout in conditions of high stress.

The COVID-19 health emergency has greatly changed the lives of professionals and parents, increasing perceptions of stress and specific burnout symptoms such as emotional exhaustion, emotional distancing, and depersonalization, as well as a decrease in feelings of satisfaction and accomplishment. Among healthcare professionals serving minors in therapeutic communities, the pandemic has required the use of numerous emotional and cognitive resources. The most obvious component was emotional exhaustion, which is consistent with what is reported in the literature; depersonalization did not experience much change, and satisfaction with one’s work was significantly lower. The constant approach to the minor with difficulties during a period of health uncertainty may have influenced the reduction in the emotional tolerance threshold, which is associated with a general deterioration in emotional abilities. Among operators, the use of avoidance strategies increased, which is understandable from the point of view of increased emotional intolerance. However, feelings of trust and solidarity may have played a protective role. The COVID-19 pandemic has upset habits and caused an obvious cost in both personal and social terms.

The results of the study confirm this view and invite reflection on the immediate impact of working with patients and, in particular, on the state of mental well-being of the healthcare professionals involved. The findings highlight the importance of providing adequate support to these types of workers and the need to proactively support them in the event of similar crises in the future. The healthcare worker is a care provider who has a strong relational and emotional closeness with the minor users of the therapeutic communities. Therefore, early intervention is necessary to prevent burnout symptoms. As is commonly known, the occurrence of psychological discomfort in the workplace can have an impact on the therapeutic relationship. Changes in emotional competencies and stress levels associated with burnout syndrome may adversely affect caregiver parenting. There is a need to explore how to respond to the pandemic so that dysfunctional adaptation can be readily identified.

## Figures and Tables

**Figure 1 ijerph-19-16320-f001:**
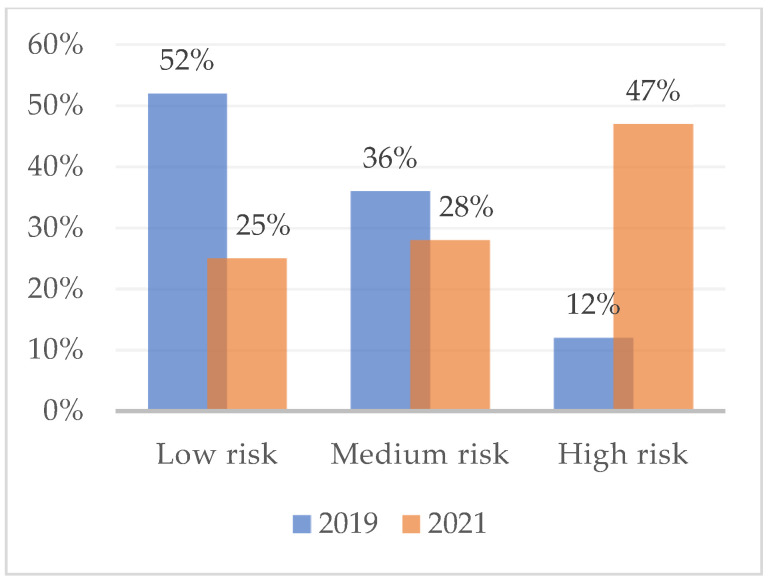
Risk for burnout syndrome due to emotional exhaustion in February 2019 and in April 2021. *n* = 75.

**Figure 2 ijerph-19-16320-f002:**
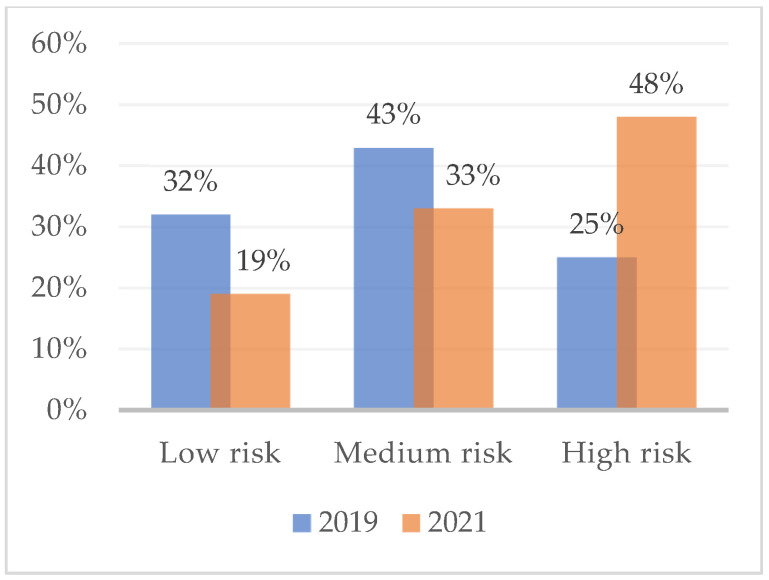
Risk for burnout syndrome due to personal accomplishments in February 2019 and in April 2021. *n* = 75.

**Figure 3 ijerph-19-16320-f003:**
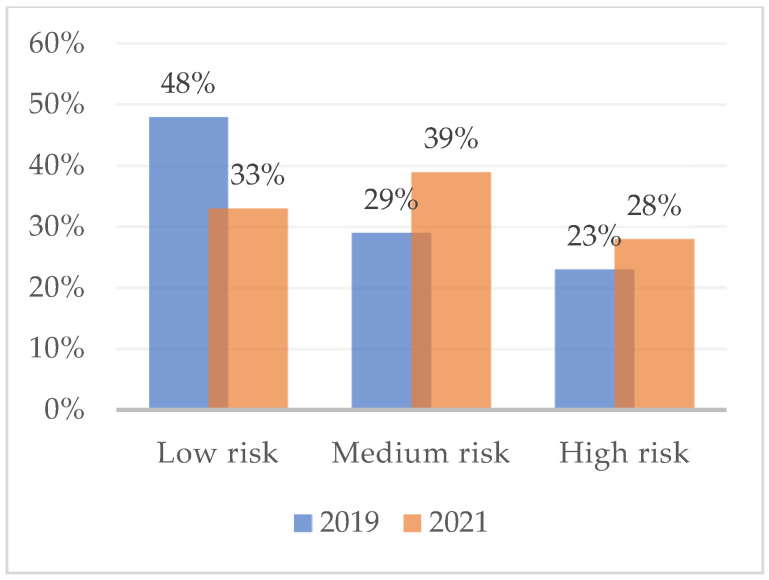
Risk for burnout syndrome due to depersonalization in February 2019 and in April 2021. *n* = 75.

**Table 1 ijerph-19-16320-t001:** Respondent profile: socio-demographic characteristics.

Characteristics	*n*	%
Gender		
Female	54	72.4%
Male	21	27.6%
Educational Level		
Middle school	1	1.3%
High school	2	2.6%
University	31	42.1%
Postgraduate degree	19	25.1%
Professional qualification	22	28.9%
Marital status		
Single	15	20%
Married	30	40%
Divorced	6	8%
Partner cohabiting	18	24%
Partner no cohabiting	6	8%
Physical activity		
Never	8	10.6
Occasionally	35	46.7%
Regularly	32	42.7%
Site		
Northern Italy	75	100%

*n* = 75. Sample in April 2021.

**Table 2 ijerph-19-16320-t002:** Professional information.

Characteristics	*n*	%
Distance from workplace		
10–11 min	23	30.6%
11–30 min	32	42.7%
+31 min	20	26.7%
Working hours		
Full time	58	77.3%
Part-time	17	22.7%
Shift worker		
Yes	45	60%
No	30	40%
Employment contract		
Permanent	65	86.6%
Fixed-term	8	10.7%
Independent	2	2.7%
Number of minors		
1–3	1	1.4%
4–7	10	13.3%
8–10	58	77.3%
+11	6	8%
Age of minors		
9–14	38	50.7%
15–18	37	49.3%

*n* = 75. Sample in April 2021.

**Table 3 ijerph-19-16320-t003:** COVID-19 survey.

Characteristics	*n*	%
Episodes of physical and verbal aggression by users		
Unchanged	27	36%
Increased	48	64%
Number of users in the structure		
Decreased	6	8%
Unchanged	41	54.7%
Increased	28	37.3%
Interaction with colleagues		
Worsened	30	40%
Unchanged	25	33.3%
Improved	20	26.7%
Quarantines		
Any	21	28%
One	20	26.7%
More than one	34	45.3%
Anger		
Decreased	17	22.8%
Unchanged	29	38.6%
Increased	29	38.6%
Trust		
Decreased	34	45.3%
Unchanged	26	34.7%
Increased	15	20%
Fear		
Decreased	19	25.3%
Unchanged	24	32%
Increased	32	42.7%
Support		
Decreased	6	8%
Unchanged	21	28%
Increased	48	64%
Concern		
Decreased	8	10.6%
Unchanged	19	25.3%
Increased	48	64.1%
Utility		
Decreased	7	9.3%
Unchanged	20	26.7%
Increased	48	64%
Anxiety		
Decreased	25	33.3%
Unchanged	21	28%
Increased	29	38.7%
Effectiveness		
Decreased	10	13.3%
Unchanged	27	36%
Increased	38	50.7%
Distress		
Decreased	27	36%
Unchanged	24	32%
Increased	24	32%
Confusion		
Decreased	28	37.3%
Unchanged	16	21.3%
Increased	31	41.4%

*n* = 75. Sample in April 2021.

**Table 4 ijerph-19-16320-t004:** Statistics on burnout syndrome.

Questionnaire	February 2019		April 2021		*z* Value	*p*
	M	SD	M	SD		
Emotional exhaustion	15.29	8.22	24.61	13.22	5.82	<0.001
Depersonalization	5.2	4.21	6.68	5.81	4.39	<0.001
Personal accomplishment	33.7	5.77	29.97	6.95	−4.78	<0.001

*n* = 75.

**Table 5 ijerph-19-16320-t005:** Statistics on psychological assessment.

Questionnaire	February 2019		April 2021		*z* Value	*p*
	M	SD	M	SD		
TAS-20						
Difficulty identifying feelings	11.83	4.36	15.11	6.92	5.35	<0.001
Difficulty describing feelings	11.8	3.99	13.35	4.36	4.16	<0.001
Externally-oriented thinking	16.63	4.58	18.27	4.99	3.32	<0.001
Total score	40.25	9.78	46.72	13.96	5.08	<0.001
IRI						
Perspective taking	17.2	4.14	19.23	4.14	3.87	<0.001
Fantasy	8.97	3.65	9.17	3.67	0.73	0.5
Empathic concern	17.03	2.98	17.73	3.16	3.02	0.003
Personal distress	10.2	4.78	8.05	3.6	−4.75	<0.001
FFMQ						
Observing	25.45	6.26	27.4	6.45	3.16	0.002
Describing	28.76	6.09	26.79	6.1	−3.34	<0.001
Acting with awareness	30.88	5.4	26.95	8.23	−5.20	<0.001
Non-judging of inner experience	28.84	5.9	26.43	6.72	−4.76	<0.001
Total score	134.81	18.43	127.53	21.34	−2.86	0.004
COPE IV						
Social support	32.6	5.77	32.52	5.77	−0.02	1
Avoidance strategies	24.73	4.29	27.3	6.5	5.1	<0.001
Positive attitude	35.83	5.42	36.76	5.7	2.41	0.01
Problem-solving	35.27	5.43	34.33	4.8	−1.09	0.3
Turning to religion	17.7	3.63	19	4.1	3.18	0.001
Total score	144.31	13.64	148.33	13.76	4.21	<0.001
FDS_R						
Discomfort intolerance	15.85	4.38	18.48	6.67	2.78	0.005
Achievement	17.6	4.48	19.57	5.31	2.74	0.006
Entitlement	19.75	4.88	22.03	6.2	2.47	0.01
Emotional intolerance	16.45	4.78	19.53	6.36	3.69	<0.001
Total score	69.65	15.47	79.61	21.87	3.55	<0.001

*n* = 75.

## Data Availability

The datasets that were generated for this study are available upon request from the corresponding author.

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
