# Peer review of "Burnout Syndrome and COVID-19 Lockdown: Research on Residential Care Workers Who Assume Parental Roles with Youths"

_ijerph, 2022, doi:10.3390/ijerph192316320_

Round 1

Reviewer 1 Report

:  Investigating the impact of COVID-19 on health professionals is a timely and clinically relevant undertaking.  This paper reflects one such undertaking. While addressing a gap in existing literature, there are areas where more information or clarification is needed.  These areas are highlighted below.

General comment: the term “anamnestic” was unfamiliar to this reader – suggest using a more common term – not certain whether it relates to demographic data or something else.

Introduction: Given the title, please consider defining terms such as “therapeutic communities”, making clear what these are; clearly define healthcare professionals early in the paper; define the population better – the title implies adolescent, but the term “children and adolescents” is employed in many places in the paper. The linkage between a health professional’s role and a parental role needs to be made more explicit.  It seems the study is based on a huge assumption for which data are not collected in this paper.

Design:  support for the design is questionable.  Typically, longitudinal studies collect data more than twice.  This really seems like a descriptive correlation study with two data collection points.  Perhaps some discussion of the initial study and how those findings contributed to the current study would be helpful.

Sample: as mentioned above, more information is needed about the sample - - three year degree does not really provide sufficient information for the reader. Including some information about inclusion and exclusion criteria would strengthen the manuscript.

Instruments: More information needed about all instruments employed in this study.  As mentioned previously, please clarify what constitutes an “anamnestic form.” In addition, please provide more information the psychometric properties of all instruments, the number of items per instrument, and how the instrument is scored. 

It might help to have very clear research questions, as a way to help the reader understand the findings better.

Data analysis:  Reference to “mediating factors” seems out of place given the type of analysis performed.  Typically one would anticipate seeing reference to SEM or some other more advanced statistical method. Analyses all seem to be bivariate in nature.  Please clarify.

Findings:  Please include sample numbers in all tables.  Reference is made to Table 5 but I could not Table 5 in the manuscript.

Conclusions:  Seem to support what is known about COVID-19 – it’s advent resulted in many challenges. 

Author Response

Thank you for this careful review.

- Regarding the term "anamnestic," we refer to an ad hoc file in which we examine sociodemographic and occupational information. We assumed to keep the term but to deepen the meaning to avoid ambiguity.

- In the introduction, we explained the meaning of the structures that are the subject of our study and the meaning of "health professionals." We also suggested changing the title to include the more general term "youth" to include both adolescents and children. We added a large paragraph to explain the "parental" functions of those who work in residential facilities for minors.

- We have thought about the term "longitudinal analysis" and understand the confusion about our decision to use it. To make our work as transparent and understandable as possible, we have eliminated this term. We have therefore focused on the data collection that took place at both time points, and thus on the presence of two distinct objectives (to this end, we have also changed the abstract slightly)

- We have included inclusion and exclusion criteria and specified the professions involved in the research

- We have better described the instruments by including specific subparagraphs

- We improved the clarity of the objectives (changes in the abstract and the last part of the introduction)

- We deleted the term "mediating factors" because our analyzes were purely correlational. We would like to analyze any mediating relationships in the future.

- We have corrected the caption of the tables. The reference to Table 5 was a typo.

Reviewer 2 Report

I appreciated the opportunity to review “Impact of Covid-19 Lockdown in Therapeutic Communities: How Did it Affect Burnout in Health Professionals who Assume Parental Roles with Vulnerable Adolescents?”. Although I have seen other recent work about "burnout" among health/ social service professionals during the COVID-19 pandemic, this manuscript takes a different perspective by focusing on those who assume parental roles with vulnerable adolescents. This is also a well-written and informative manuscript. Still, it would be helpful to provide more background information.

1.     There is a need to explain the sampling method and justify how the collected data can offer valuable information. There may be a need to further demonstrate the methodological rigor.

2.     Brief of the research site’s background is needed. There are different stages of the COVID pandemic – from the 'acute phase’ to 'less severe phase’. It may be helpful to state briefly when the data was collected and how the lockdown affected people's daily life.

3.     Please give examples to explain what “health professionals” are. Explain briefly what “adolescents in vulnerable conditions” means. Again, examples are helpful.

4.     Could you identify any gender differences from the findings?  (See for example Purvanova & Muros, 2010).

5.     Is it possible to consider the changes/difficulties they experienced during the outbreak as an issue of “adaptation” or “adjustment” (See Ling, Shum, Kwan & Xu, 2021).

6.     Explain briefly why the study may interest the readers of this international academic journal, given the study was conducted in a very unique context (i.e., northern Italy).

7.     Practical implications proposed in the article may be too brief.

8.     Some minor format issues need to be addressed.

I enjoyed reading this manuscript. I wish the authors well. Thanks. 

Author Response

Thank you for your careful review.

  1. We have reviewed the paragraph that refers to the recruitment of the sample and the methodology applied (inclusion and exclusion criteria)
  2. We have included in the introduction a more detailed explanation of the period during which the data were collected (what limitations there were) and the impact on workers.
  3. We have also explained these terms in the introduction. We also indicated the occupations included in the sample.
  4. We performed the analysis but did not find any differences. We have added a sentence in the results to explain this.
  5. Thank you for the suggestion! We have thought about it and looked at the proposed article. So in the discussions we refer to the possibility that workers had to adapt to a new stressful situation
  6. We are aware that the sample so composed may be a limit. In the introduction, we explain how this unique relationship between a health professional and a youth can characterize the different settings present at the international level.
  7. We have integrated the paragraph by deepening the practical perspectives.
  8. In this point, we have some correction difficulties because the adaptation to the template prevents us from making the structure correct.